# Effects of Using an Alternative Bedding Composition on the Levels of Indicator Microorganisms and Mammary Health in Dairy Farm Conditions

**František Zigo [1,*](#), Naďa Sasáková [2], Gabriela Gregová [2](#), Jana Výrostková [3] and Silvia Ondrašovičová [4]**

[1] Department of Animal Husbandry, University of Veterinary Medicine and Pharmacy, Komenského 73, 04181 Košice, Slovakia

[2] Veterinary Legislation and Economy, Department of the Environment, University of Veterinary Medicine and Pharmacy, Komenského 73, 04181 Košice, Slovakia; nada.sasakova@uvlf.sk (N.S.); gabriela.gregova@uvlf.sk (G.G.)

[3] Department of Milk Hygiene, University of Veterinary Medicine and Pharmacy, Komenského 73, 04181 Košice, Slovakia; jana.vyrostkova@uvlf.sk

[4] Department of Physiology, University of Veterinary Medicine and Pharmacy, Komenského 73, 04181 Košice, Slovakia; silvia.ondrasovicova@uvlf.sk

* Correspondence: frantisek.zigo@uvlf.sk; Tel.: +421-908-689-722

**Abstract:** The aim of this study was to compare an improved bedding composition with conventional straw bedding under farm conditions, regarding its effects on the influence of indicator microorganisms on the hygiene levels of cubicle floors and the occurrence of mastitis in dairy cows. Dairy cows were housed in newly built stalls divided into two parts, each with four subsections, and bedded cubicles arranged in three rows. Five stall subsections from each 9-bedded cubicle were selected for study, and 30 dairy cows were monitored according to the time intervals of bedding treatment for cubicles. In the first subsection (control), the cows were housed in bedded cubicles layered with straw up to a height of 20 cm. Sections 2–5 had alternative bedding (AB) as follows: fresh AB, AB 1 month old, AB 2 months old, and AB 3 months old, which were bedded one day before (fresh) and 1–3 months before the actual observation period, respectively. The alternative bedding per one cubicle consisted of ground limestone (100 kg), water (80 L), recycled manure solids (RMS; 15 kg), and straw (25 kg). After laying, the bedding was treated with a concrete selector to provide strength and sufficient resistance. A total of 180 bedding and 600 quarter milk samples were taken simultaneously from all five monitored subsections for microbiological determination. Comparing classical straw bedding with the alternate bedding showed a stabilizing effect by keeping the bedding thickness up to the floor barrier level, which had a beneficial effect by reducing the level of fecal contamination in the rear of the cubicle. Fecal coliforms and fecal streptococci were found to be reduced in one-day-old bedding as well as after the first, second, and third months. By evaluating the health status of the mammary glands, a positive effect was noted in reducing the occurrence of subclinical mastitis, which was reflected in a reduced number of infected quarters in the group of cows housed in cubicles for three months after use of improved bedding.

**Keywords:** dairy cows; hygiene; bedded cubicles; recycled manure solids; microorganisms

## 1. Introduction

Ensuring optimal conditions for animal welfare and production potential is one of the most common problems for breeders and has been a debated topic for researchers for a long time. Technological

animal husbandry systems are one of the relevant indicators. The ideal conditions for any animal are in a system of housing that is satisfactory in terms of active health production, allowing normal manifestations of behavior, as well as providing a high standard of health care [1,2]. The main influences on the hazards and risks associated with aspects of housing and its management are illustrated in Figure 1 [3].

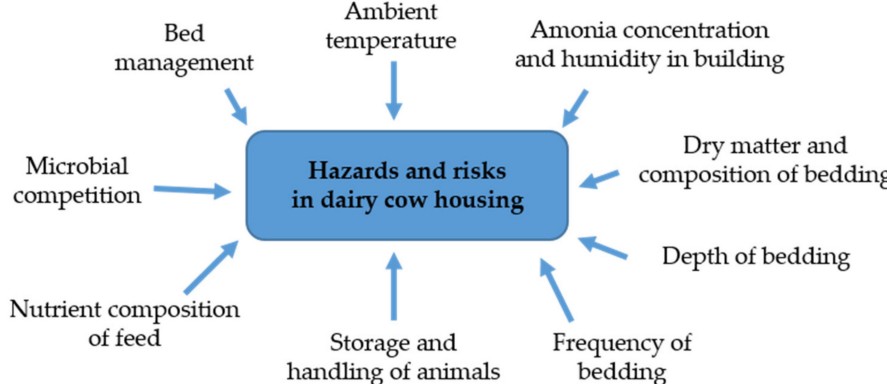

**Figure 1.** Factors affecting hazards and risks associated with bedding materials in dairy cow housing. Sources: modified figure according to Bradley et al. [3].

Animal housing is the basis of the technological system. Its method determines the choice of other parts of the technological system. Housing parameters should create optimum conditions so that potent biological material, supplied with full nutrition, can realize its production capabilities [4].

In particular, the space requirements of animals with regard to their natural needs must be respected. Housing parameters must be chosen in such a way that the animals are kept clean without a great need for manual work. Long-term maintenance with optimal parameters to ensure cleanliness in housing is a common problem for most dairies with marked milk production. Deteriorating hygienic housing conditions and the daily routine of milking cows can affect the contamination and quality of the produced milk because it increases the potential transfer of microorganisms from bedding to the mammary gland or milk [2,5].

Different housing systems have been introduced in practice for individual cattle categories, such as free housing with bedded cubicles [6] or free group pen stalls [7]. In the free group housing model, these are only two-space pens (separated by feeding and laying areas) with different types of floors and bedding. In the housing model with cubicles, there are different variants of free stalling, exclusively with bedded boxes [6,7].

Free stalling with bedded cubicles is mainly used in dairy farming. It can be successfully used for breeding heifers as well. It keeps the animals clean, which is especially important for dairy cows. It provides them with enough comfort to relax and minimizes mutual distractions between animals. The stated advantages of a bedded cubicle are only achieved if the dimensions are correctly selected according to the body size of the cattle. Adequate space must be created in bedded cubicles, not only for comfortable lying and standing but also for getting up and lying down. Otherwise, if the cubicles do not provide sufficient comfort to the dairy cows, they look for another place to lie, for example in the feeding area or in the manure corridor. In this case, the cows become excessively dirty, and this increases the risk of mammary gland inflammation [8] or claw diseases [9].

For an intramammary infection (IMI) to occur, it is necessary for the teat skin to be contaminated with pathogens, the pathogens to penetrate the teat duct, and the infection to be established in the sinuses, ducts, or tissues of the udder. With inflammation follows an increase in the level of white blood cells, and this causes an increase in the somatic cell count (SCC) of the milk [10].

In particular, contamination of the skin and udder with fecal bacteria, but also with staphylococci and streptococci from dirty bedding after milking, increases the risk of mastitis in cows. Due to time

and economic restraints, breeders do not change the entire content of the bedding in cubicles but only the part that is most contaminated [11].

In our conditions, straw is the predominant bedding material [2], although other material may be used so long as it ensures adequate hygiene and welfare of the animals [12]. Dairies are looking for alternative bedding sources, and some have implemented recycled manure solids (RMS) as bedding. Due to a lack of straw, recycled manure solids (RMS) have been used as a substitute bedding material in recent years to create sufficient comfort for dairy cows. RMS consists of dry matter and a nutrient-rich fraction obtained by mechanical or gravitational separation of slurry manure removed from dairy cow housing systems [5,12]. To ensure its hygienic quality and optimum pH, RMS is often combined with straw and other components such as limestone or zeolite [2,13].

According to available studies [2,3,5,12,13], the use of manure as an alternative source of bedding in combination with other components such as limestone or zeolite should ensure hygienic quality and optimum pH in bedded cubicles for dairy cows. The use of an alternative bedding composition is expected to ensure animal welfare as well as to reduce the infectious pressure of bacterial pathogens from the environment causing mastitis in dairy cows.

The aim of this study was to compare the effect of using an improved composition of bedding in dairy farm conditions on indicator microorganisms, with regard to the level of hygiene on cubicle floors and occurrence of mastitis in dairy cows, compared to that of conventional straw bedding.

## 2. Materials and Methods

### 2.1. Cows and Housing

The practical part of the study was carried out on a farm of 300 Slovak Spotted Cattle dairy cows in the district of Stara L'ubovňa in eastern Slovakia. The dairy cows were housed in a newly built high-air stall divided into two main sections, between which there was a feeding table (Figure 2). Each section was divided into four subsections. In each subsection there were 42 cubicles, which were arranged opposite to each other in three-rows (Figure 3). Between the rows there were movement areas with automated excrement cleaning, into which dairy cows could be moved as needed to perform technological tasks.

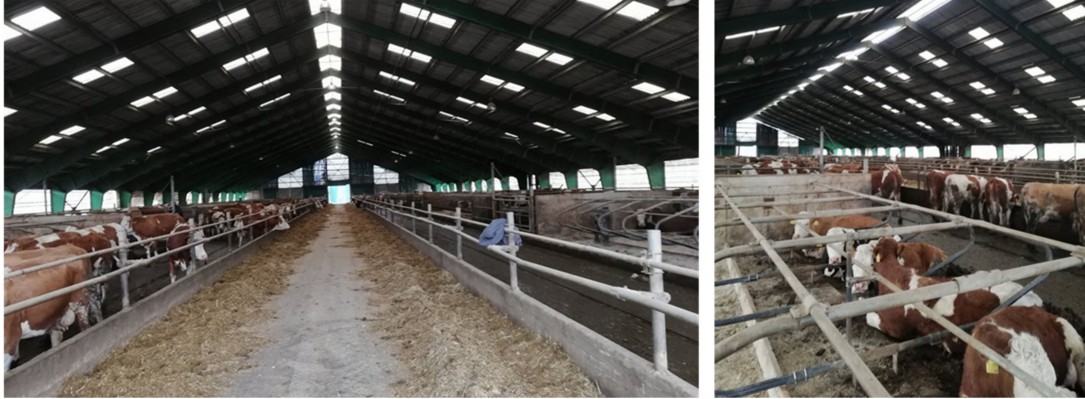

**Figure 2.** From left: Stall with feed space and cubicle beds. Photo by Zigo F. (2019).

### 2.2. Milking Cows and Milking Procedures

Cows from all sections were milked twice a day at 5 a.m. and 4 p.m. in a parallel milking parlor $2 \times 12$ (BouMatic). After loading the cows, the udder and teats were disinfected with G-Mix Power (Agromont, SR), which is composed of chlorine dioxide, and cleaned for at least 10–20 s. The cleaning was done by wiping the udder with a disposable paper towel. After cleaning, the first milk from each quarter was hand-drawn into a dark-bottomed pot, and the milk was subjected to sensory analysis. During the milking process, the pulsation ratio was 60:40 at a rate of 52 c·min$^{-1}$, and milking was

automatically terminated when the milk flow dropped to 0.2 L·min$^{-1}$. After milking, the teats were disinfected with a second disinfection Power Blue Mix (Agromont, Slovakia), which contains the active substance lactic acid. At the end of the milking process, cows were returned to their own subsections, and the milk was stored in a tank at a temperature between 4 and 6 °C.

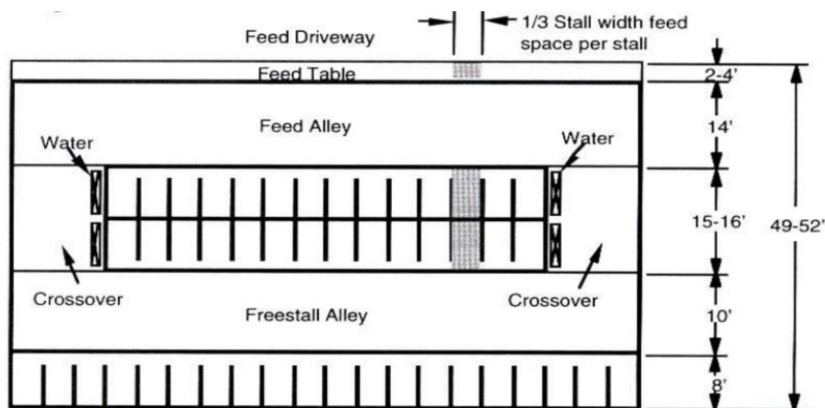

**Figure 3.** Housing module: schematic representation of one section in the stall. Sources: Samer [7].

### 2.3. Bedded Cubicles

The cubicle comprises the body space used by the dairy cow for standing and lying as well as the head space required to move the head when standing up and lying down. All cubicles in each section met the following technological requirements. For dairy cows, a comfort cubicle 2517 mm long with an active bed area of 1784 mm was used, which was delimited by a 200 mm raised board barrier above the floor level. The width of the cubicles was 1200 mm on the axis of the side barriers. The side barriers defined the space of adjacent cubicles, and their sizing affected the comfort of the animals. The height of the side barriers was 1100 mm above the box floor. At this height, a withers barrier was installed, the shape of which guides or allows movement of the animal in the lateral direction.

### 2.4. Production and Dosing of the Bedding with Improved Composition

The following components were required to produce a new type of bedding for cubicles (the components were recalculated per one cubicle): ground limestone (particle size 1–4 mm), 100 kg; water, 80 L; straw (particle size 10–15 cm), 25 kg; and recycled manure solids, 15 kg. Separated and recycled manure solids were allowed to stand for two weeks before incorporation into bedding (Figure 4). For perfect mixing, the individual components were mixed for 50–60 min in an old feeding car that was discarded for this purpose. Straw of size 10–15 cm was first introduced into the mixing compartment of the vehicle, which was then stirred for 10–15 min. Subsequently, limestone was added with the recycled manure solid. After another 20 min of stirring, the resulting mixture was diluted with water for 20–25 min (Figure 5). Immediately after mixing, the resulting slurry was dosed from the feeding car to the cleaned cubicles and spread evenly. The slurred bedding mixture was layered in two stages. In the first stage, the mixture was compacted with concrete vibrators to the desired height of 200 mm at the level of the floor barrier, which represents the thickness of the bedding. Subsequently, in the second stage, another layer (2–4 cm) of the prepared mixture was poured onto the compacted area, which was not treated with a concrete vibrator but only gently shacked with rakes (Figure 6).

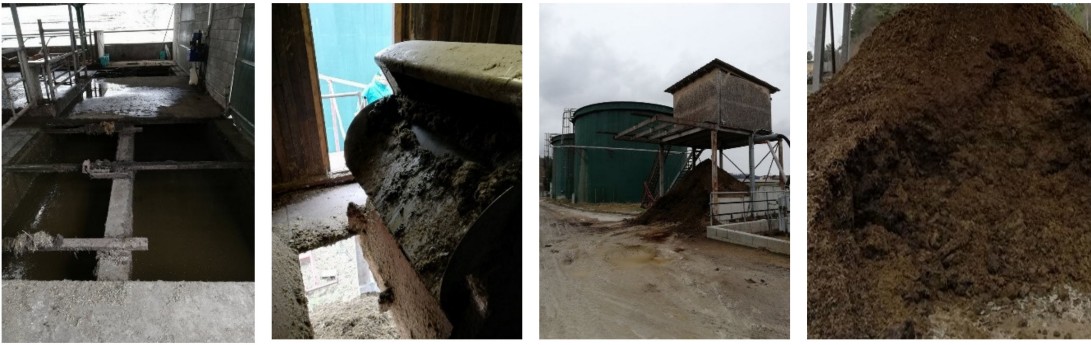

**Figure 4.** Production of recycled manure solids for bedding. From left: Manure collection in stall, solid liquid separation, fresh separated manure soil, recycled manure soil. Photo by Zigo F. (2019).

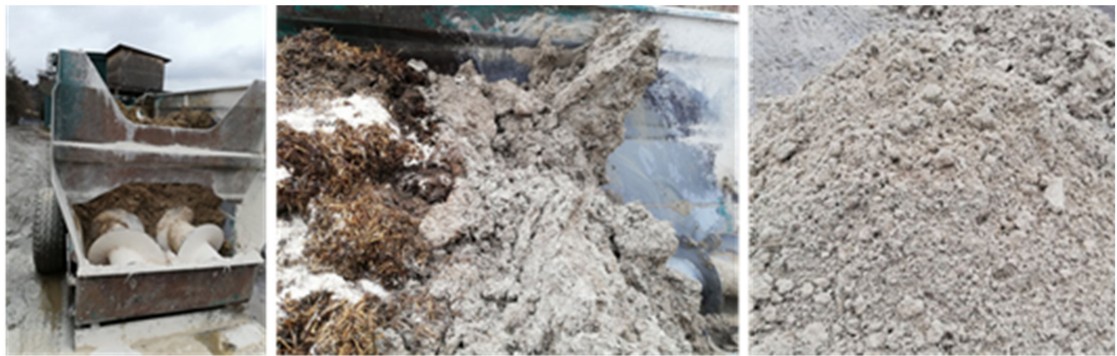

**Figure 5.** Production of bedding with improved composition for cows. Photo by Zigo F. (2019).

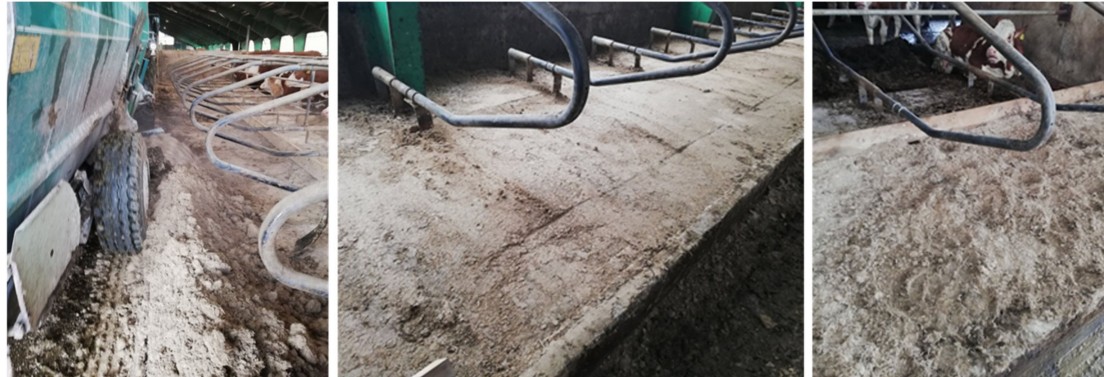

**Figure 6.** Dosing of the bedding with improved composition and its final adaptation to the cubicles. Photo by Zigo F. (2019).

*2.5. Experimental Groups Selection*

The practical part of the study was carried out on a dairy farm according to the time intervals of bedding treatment divided into individual subsections for three months before the actual samples were collected. To compare the hygienic effect on of the bedding with improved composition on the indicator microorganisms and its impact on the health of the mammary gland, five selected subsections from the stall were monitored simultaneously. From each subsection, nine cubicles (three cubicles in one row) and 30 dairy cows, in the lactation interval of at least one month after calving and one month before drying, were monitored. Four subsections with alternative bedding—AB fresh, AB after 1 month, AB after 2 months, and AB after 3 months—were bedded one day before (fresh) and 1–3 months before the actual observation period, respectively. In the fifth, control subsection (C-straw), the cows were housed in the bedded cubicles layered with straw up to a height of 20 cm. Once a day, soiled bedding was removed from the back of control cubicles and from passages, and the cubicles

were always re-bedded with fresh straw after the manure was cleaned out from the end of the bed and from the corridors.

## 2.6. Sampling of Bedded Cubicles

A total of 180 bedding samples for microbiological examination were taken simultaneously from all five subsections and monitored cows. Bedding samples were taken from four subsections according to a new schedule in intervals of one day and 1–3 months after use of alternative bedding. A control sample of bedding, consisting of straw, was taken from the last, fifth section. For each monitored subsection, nine cubicles were selected and 36 bedding samples were collected. The selection of sampling cubicles consisted of their placement in subsections. In each subsection, = three cubicles located at the beginning, middle, and end of each row were monitored. Two samples to a depth of 10 cm were taken from the middle and two from the back part of each selected cubicle. The samples of bedding were mixed and ground to a particle size of 2–4 mm prior to analysis. In addition to sampling, the depth of the bedding was measured. From the selected cubicles of each row in the monitored subsections, the height of bedding in the middle and back parts of the lying area was measured using a sliding meter, and mean values in cm were compared in Figure 9.

## 2.7. Udder Health Examination and Milk Sampling

Udder health was evaluated, and 600 quarter milk samples were taken simultaneously from all selected cows for detection of bacterial pathogens causing mastitis. From each subsection, mammary health and occurrence of mastitis of 30 dairy cows were monitored in the lactation interval of at least one month after calving and one month before drying. A thorough evaluation of udder health included clinical examination, sensory analysis of milk from forestripping of each udder quarter, followed by assessment of the California mastitis test (CMT) (Indirect Diagnostic Test, Krause, Denmark). Milk from every quarter was mixed with the reagent, and the result was scored as negative, trace, or positive (score 1–3) depending on the formation of gel in the milk sample according to Jackson and Cockroft [14]. Next, milk samples from each quarter were aseptically collected for bacteriological cultivation in accordance with the guidelines of the National Mastitis Council [15]. The samples were cooled to 4 °C and immediately transported to the laboratory and analyzed on the following day.

## 2.8. Laboratory Analyses

### 2.8.1. Microbiological Examination of Bedding Samples

Ground bedding samples of 10 g were taken, and 90 mL of saline was added. Following subsequent homogenization, individual ten-fold dilutions by watering (1 mL) in physiological saline were then prepared from the treated samples. Dilutions from $10^{-4}$ to $10^{-8}$ according to the methods of Fournel et al. [5,12] were used for seeding. For the surface of the culture media thus prepared, the spreading method (0.1 mL) was used for the individual microorganisms. Various culture media were used to ensure appropriate growth conditions for the individual microorganism strains tested. For total count of bacteria (total viable count, TVC) at 37 °C, Plate Count Agar (Oxoid, UK) was used. Fecal coliform bacteria (FCB) were cultivated on a petri dish containing McConkey agar (Oxoid, UK) at 43 °C. For coliform bacteria (CB) at 37 °C, Endo agar (Hi-Media, India) was used. For fecal streptococci (FS) at 37 °C, M17 agar (Oxoid, UK) was used. The culture media were prepared according to the manufacturer's instructions and poured into 90 mm diameter Petri dishes in parallel. After a specified incubation time for each microorganism, the colony forming units (CFUs) were calculated according to the appropriate formula. Reported results were (mean ± standard deviation) calculated as the average of two replicates and log-transformed.

### 2.8.2. Microbiological Examination of Milk Samples

Bacteriological examinations were performed according to commonly accepted rules by Malinowski et al. [16]. Quarter milk samples (10 μL) were cultured at the respective veterinary practice according to routine procedures, usually employing Columbia Blood Agar Base with 5% of defibrinated blood, Staphylococcal medium N° 110, Baird-Parker agar, Edwards Medium, and Mac Conkey Agar (Oxoid, OXOID Ltd., Basingstoke, Hants, UK) and incubated at 37 °C for 24 h. As well as evaluating bacterial growth characteristics, other assays were used to identify bacterial species: pigment and coagulase production, catalase activity, hemolysis, Gram staining, and other virulence factors. *Staphylococcus* spp. were selected for the tube coagulase test (Staphylo PK, ImunaPharm, Šarišské Michal'any, Slovakia). Suspected colonies of *Staphylococcus* spp., *Streptococcus* spp., and *Enterobacteriacae* spp. were isolated on blood agar, cultivated at 37 °C for 24 h, and identified biochemically using the Staphy test, Strepto test, and resp. Entero test using the software TNW Pro 7.0 (Erba-Lachema, Brno-Řečkovice a Mokrá Hora, Czech Republic) according to the manufacturer's instructions.

### 2.9. Statistical Analyses

For statistical comparison, bacterial counts were also log-transformed and expressed in log $CFU.mL^{-1}$ according to Rychtáriková et Kuncová [17]. The differences between tested indicator microorganisms as well as the heights of improved bedding and control bedding with straw were analyzed by using analysis of variance (ANOVA) followed by Dunnett's multiple range test. The minimum criteria for statistical significance was set at $p \leq 0.05$ for all. The differences in the CMT score, prevalence of mastitis, and distribution of bacterial pathogens among monitored groups of cows were statistically analyzed using the Chi-squared test. The dependence of the individual signs was tested at a significance level $\alpha = 0.05$, with critical value = 5.991.

## 3. Results

Figure 7 compares the results of the hygienic quality assessment of the bedding in the monitored subsections on the TVC and CB indicator microorganisms. Comparing cubicles from the control subsection with conventional straw bedding and cubicles with the improved recipe, bedding showed reduced counts of FCB and FS in one-day-old alternative bedding, as well as after the first and second months of its use. After three months of using the bedding with the improved recipe, no differences were detected on the indicator microorganisms TVC and CB between the compared beddings.

A similar reduction in the counts of FCB and FS indicator microorganisms was observed in the bedding samples from all subsections with the improved formulation. From the laying of this bedding (one-day-old bedding) and after the first, second, and third months of its use, reduced counts of FCB and FC were recorded compared to the conventional straw (Figure 8).

At the beginning of the study, all cubicles from the selected subsections were cleaned and filled up to the cubicle floor barrier level (200 mm) with relevant bedding. During the observation, soiled bedding was removed from the back of the control cubicles, and the cubicles were re-bedded once a day with fresh straw. Cubicles with alternative bedding were also filled up to the floor barrier level and cleaned and re-bedded once a day. Measurements of the bedding height in cubicles showed differences in the thickness among monitored subsections. The lowest mean thickness (14 cm) was recorded in the control cubicles. The optimal bedding thickness was recorded in cubicles with alternative bedding and ranged from 21 to 17 cm. Comparison of the bedding height in cubicles showed a stabilizing effect for the alternative composition of bedding, as the height of the alternative bedding was maintained at the level of the bedding barrier in one-day-old bedding as well after the first and second months of its use (Figure 9).

In addition to monitoring the composition and hygienic quality of the bedding, the health status of the mammary gland and the occurrence of mastitis were monitored in housed dairy cows in individual subsections. Table 1 describes the examination of mammary health and evaluation of the California

mastitis test (CMT) in the monitored groups of cows housed on classic straw bedding and bedding with the improved recipe. After three months of using the alternative bedding, an increased number of healthy quarters and a reduced incidence of infectious quarters were recorded. In two groups of cows housed in cubicles after two and three months of alternative bedding use, more quarters with negative CMT scores were recorded than the control group with classic straw bedding. In addition, in the group of cows housed in cubicles for three months with the improved bedding, reduced incidence of subclinical mastitis was observed (Figure 10).

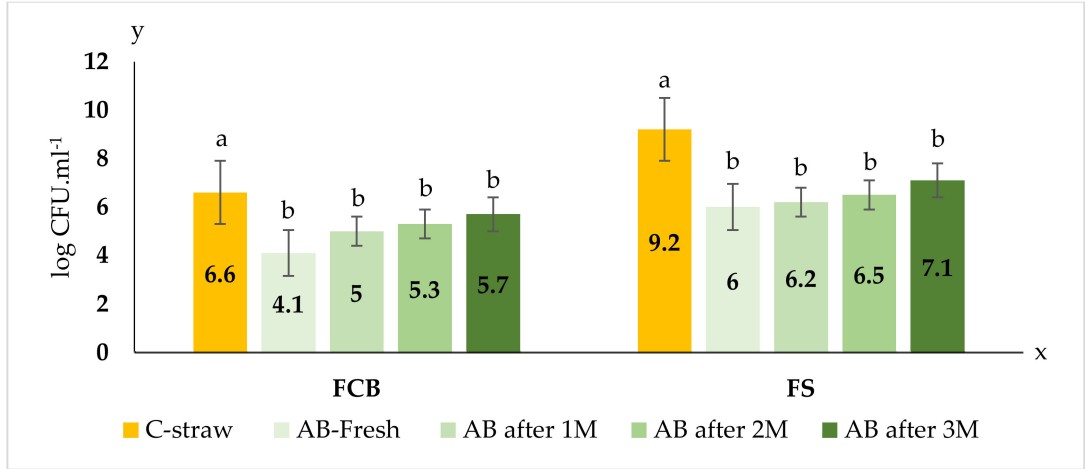

**Figure 7.** Comparison of improved bedding and straw bedding on indicator microorganisms (fecal coliform bacteria, FCB, and fecal streptococci, FS) regarding the level of hygiene. Note: C straw—control cubicles were not emptied of straw bedding, but the cubicles were re-bedded once a day with fresh straw; AB-Fresh—cubicles bedded with one-day-old (fresh) alternative bedding; AB after 1M—one month after use of alternative bedding; AB after 2M—two months after use of alternative bedding; AB after 3M—three months after use of alternative bedding; FCB—fecal coliform bacteria at 43 °C; FS—fecal streptococci at 37 °C; a,b values above the column with different superscript letters differ significantly at $p < 0.05$.

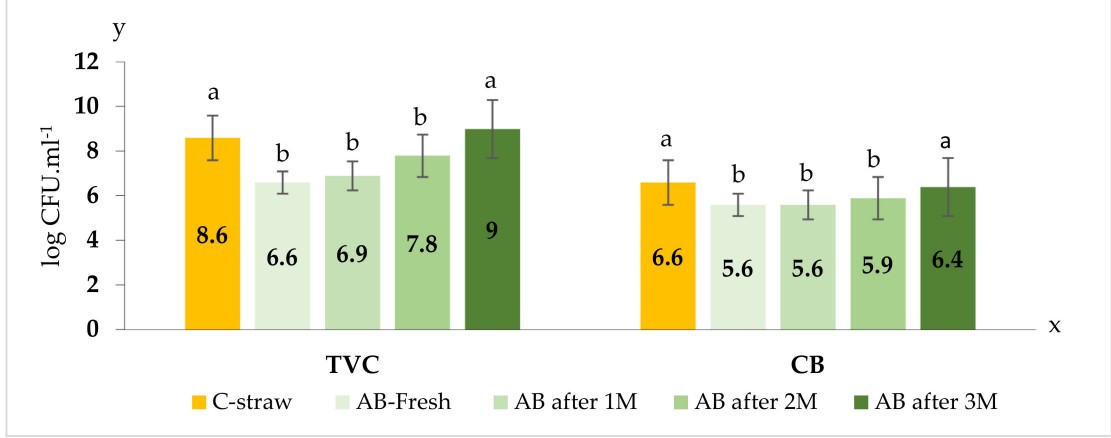

**Figure 8.** Comparison of improved bedding and straw bedding on indicator microorganisms (total viable count, TVC, and coliform bacteria, CB) regarding the level of hygiene. Note: C straw—control cubicles were not emptied of straw bedding, but the cubicles were re-bedded once a day with fresh straw; AB-Fresh—cubicles bedded with one day old (fresh) alternative bedding; AB after 1M—one month after use of alternative bedding; AB after 2M—two months after use of alternative bedding; AB after 3M—three months after use of alternative bedding; TVC—total viable count at 37 °C; CB—coliform bacteria at 37 °C; a,b values above the column with different superscript letters differ significantly at $p < 0.05$.

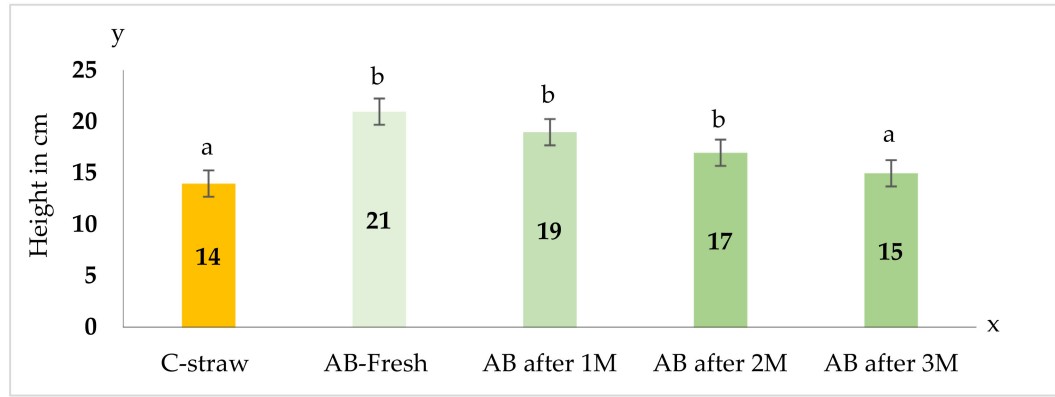

**Figure 9.** Comparison of bedding heights in cubicles. Note: C fresh straw—control cubicles were completely emptied of old straw bedding and layered with fresh straw; C straw—control cubicles were not emptied of straw bedding, but the cubicles were re-bedded once a day with fresh straw; AB-Fresh—cubicles bedded with one-day-old (fresh) alternative bedding; AB after 1M—one month after use of alternative bedding; AB after 2M—two months after use of alternative bedding; AB after 3M—three months after use of alternative bedding; [a,b] values above the column with different superscript letters differ significantly at *p* < 0.05.

**Table 1.** Examination and evaluation of the California mastitis test (CMT) in monitored groups of cows.

| Cows/CMT | | C-Straw | AB-Fresh | AB After 1M | AB After 2M | AB After 3M |
|---|---|---|---|---|---|---|
| Number of animals in group | | 30 | 30 | 30 | 30 | 30 |
| Number of examined quarters | | 118 | 116 | 119 | 120 | 118 |
| Healthy quarters [1] (%) | | 79.2 [a] | 78,4 | 80.6 | 83.4 | 86.4 [b] |
| Infected quarters [2] (%) | | 15.2 [a] | 16.3 | 13.4 | 11.6 | 9.3 [b] |
| CMT score and correlation with SCC of examined quarters | | | | | | |
| CMT score | SCC [3] $\times 10^3$ | % | % | % | % | % |
| *n* (negative) | 0–200 | 56.6 [a] | 57.7 | 62.2 | 66.7 [b] | 72.8 [b] |
| T (trace) | 200–400 | 22.6 [a] | 20.7 | 18.4 | 16.7 | 13.6 [b] |
| 1 | 400–1.200 | 11.0 | 9.5 | 8.4 | 7.5 | 5.9 |
| 2 | 1.200–5.000 | 5.7 | 6.9 | 5.9 | 5.0 | 4.2 |
| 3 | over 5.000 | 4.2 | 5.2 | 5.0 | 4.2 | 3.4 |

Note: Healthy quarters [1]—quarters with negative or trace CMT scores, no change in milk consistency, and no clinical signs of mastitis; Infected quarters [2]—positive quarters after bacteriological investigation (see Table 2). SCC [3]—Somatic cell count correlation with CMT score was performed according to Jackson and Cockcroft [14]; C-straw—control group of cows housed in cubicles that were not emptied of straw bedding, but the cubicles were re-bedded once a day with fresh straw; AB-Fresh—cubicles bedded with one-day-old (fresh) alternative bedding; AB after 1M—one month after use of alternative bedding; AB after 2M—two months after use of alternative bedding; AB after 3M—three months after use of alternative bedding; [a,b] Different superscript letters in rows between tested groups with alternative bedding and control indicate significant differences when level α = 0.05 (5%); critical value $\chi^2 = 5.99$.

Quarter milk samples were taken from all selected cows in the monitored subsections and analyzed for bacterial pathogens causing mastitis. Table 2 shows representations of the most common bacteria causing subclinical and clinical mastitis in the selected groups. In all groups from quarter milk samples, coagulase-negative staphylococci, *S. aureus*, and streptococci were isolated. In particular, *S. aureus* was isolated from clinical forms of mastitis together with *S. chromogenes* and *Str. uberis.* By using bedding with alternative composition, a reduced number of infected quarters was demonstrated after three months. Reduction in the incidence of subclinical mastitis mainly was due to a reduction in intramammary infections caused by *E. coli* and mixed infections.

**Table 2.** Isolated microorganisms from infected quarters in monitored groups.

| Monitored Groups | Infected Quarters | | Subclinical [1] | | | | | Clinical [2] | | |
|---|---|---|---|---|---|---|---|---|---|---|
| | | | CNS [3] | *S. aureus* | *Str.* spp. [4] | *E. coli* | Mix. inf. [5] | CNS [3] | *S. aureus* | *Str.* spp. [4] |
| | *n* | % | % | % | % | % | % | % | % | % |
| C-straw | 18 | 15.4 [a] | 3.4 | 1.8 | 0.8 | 1.8 | 2.5 | 0.8 | 2.5 | 1.8 |
| IB-fresh | 19 | 16.3 | 4.3 | 1.7 | 1.7 | 1.7 | 1.7 | 2.6 | 1.7 | 0.9 |
| IB after 1 month | 16 | 13.4 | 4.2 | 0.8 | 1.7 | 0.8 | 0.8 | 1.7 | 2.6 | 0.8 |
| IB after 2 months | 14 | 11.7 | 3.3 | 1.7 | 2.6 | | | | 3.3 | 0.8 |
| IB after 3 months | 10 | 8.5 [b] | 3.4 | | 1.7 | | | | 2.5 | 0.8 |

Note: *n*—number of isolated bacteria from examined quarters in each group; Subclinical mastitis [2]—no signs were observed, the udder and milk appeared normal, but infection was still present with positive CMT score and increased SCC; Clinical mastitis [3]—signs ranged from mild to severe with positive CMT score, high level of SCC, positive bacteriological cultivation, changing the consistency of the milk with the presence of flakes, clots, or pus, and reduction or loss of milk production with clinical signs; CNS—coagulase-negative staphylococci (*S. chromogenes*, *S. epidermidis* and *S. warneri*); Str. spp. [4]—*Streptococcus sanguinis* and *Streptococcus uberis*; Mix. inf. [5]—mixed infection caused by two or more bacteria. [a,b] Different superscript letters in rows between tested groups with alternative bedding and control indicate significant differences when the level $\alpha = 0.05$ (5%); critical value $\chi 2 = 5.99$.

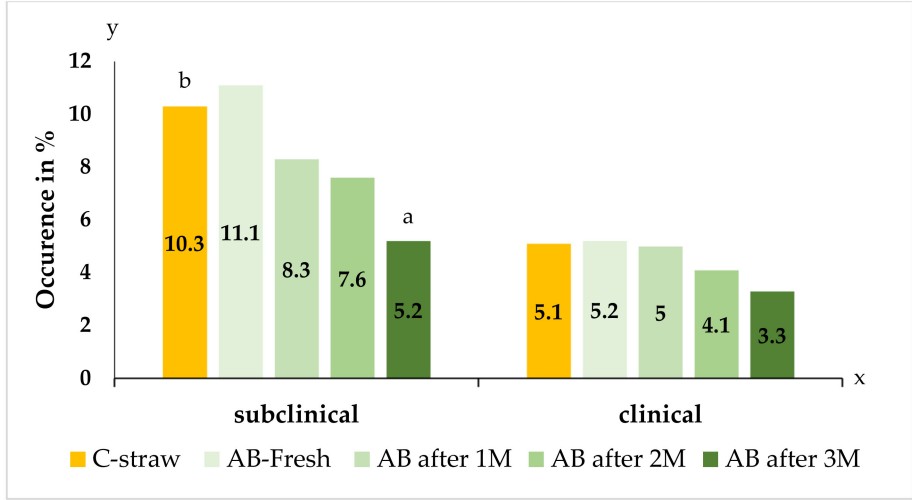

**Figure 10.** Comparison of individual forms of mastitis in the monitored groups. Note: C-straw—control group of cows housed in cubicles that were not emptied of straw bedding, but the cubicles were re-bedded once a day with fresh straw; AB-Fresh—cubicles bedded with one-day-old (fresh) alternative bedding; AB after 1M—one month after use of alternative bedding; AB after 2M—two months after use of alternative bedding; AB after 3M—three months after use of alternative bedding; [a,b] values above the column with different superscript letters differ significantly when the level $\alpha = 0.05$ (5%); critical value $\chi 2 = 5.99$.

## 4. Discussion

There are a number of cubicle features that can affect welfare. Ideally, a cubicle will allow an animal to lie down and rest without colliding or rubbing against partitions. If comfortable cubicle base and bedding, along with the correct cubicle dimensions, are used, then cows will be encouraged to spend time lying down. This will have a direct bearing on rumination, the condition of their feet, and the incidence of mastitis. Constructing a cubicle bed from concrete is a common practice. However, bare concrete is not an acceptable surface, and to ensure cows spend time lying in the cubicle, it should be covered with bedding [2,18].

Proper bedding selection and application of appropriate components to achieve a disinfectant effect is one of the key factors in maintaining cow health. Disinfection effectiveness affects the resistance to microorganisms, the selection and use of disinfectant components, and the external environment in which the disinfection process takes place [19,20].

Bedded cubicles either have a flat, raised floor or a deepened base above the dunging channel. Soft bedded cubicles with non-slip floors and good insulating properties are preferred in order for cows to lie comfortably. Loose bedding fulfils these requirements. In our conditions, straw was predominantly used for bedding, although other materials may also be used [2,21].

Recently, the separated solid fraction of RMS has started being applied to bedding in cubicles. With a higher dry matter content, it is a suitable material for lining deepened bedded cubicles. However, freshly separated manure solid has a dry matter content of only about 30%, and it is desirable to dry it before use in bedding [5,11].

To increase the hygienic quality of RMS, it is better to mix it with ground limestone in a ratio of 3(RMS):1(limestone) before applying it; however, it may be used occasionally without mixing with limestone [2]. In our study we used RMS that was left standing for two weeks for the production of bedding. The RMS was mixed with limestone at a ratio of 1(RMS):4(limestone) to increase the proportion of limestone and the accompanying disinfectant effect of the bedding. The increased disinfection effect was confirmed from the taken samples. Comparing classical straw bedding with the improved bedding, we found that the total viable count and coliform bacteria were reduced in one-day-old alternative bedding as well as after two months of its use. In addition to TVC and CB, decreased numbers of fecal coliform bacteria and fecal streptococci were also observed in one-day-old bedding as well as after the first, second, and third months of use.

According to Tančin et al. [21], the bedding thickness in the cubicle should be at least 150 mm to create a well-formable bed with good insulation. Relatively speaking, the best bed is formed in so-called deepened-base cubicles. The bed is confined by a barrier above the floor level and chest plate. The floor barrier level prevents the bedding from being expelled from the cubicle, so there is a decreased need to supplement the bedding. At the beginning of our study, all cubicles were filled with bedding up to the floor barrier level (200 mm). The floor barrier level defines the width and height of the active bed area for dairy cows in which the bedding is situated. Comparison of the bedded cubicles showed the stabilizing effect of keeping the bedding thickness up to the floor barrier level in the cubicles with one-day-old alternative bedding and after two months of use compared to the control cubicles filled with conventional straw.

According to Brouček et al. [13], the top of the bedding barrier should be at the floor level of the cubicle, i.e., 200 mm above the movement corridor, with a width of 100 mm. As already mentioned, the floor barrier is supposed to reduce bedding loss from the cubicle, as well as the need to supplement the bedding in it, but with traditional straw bedding it is usually necessary to add 2–3 kg of straw per day to each cubicle so that the bedding floor stays flush with the top of the floor barrier level.

It often happens that, although in a straw-bedded cubicle there is initially sufficient bedding height, the cows by their own weight spread the bedding to the sides when lying down and getting up. When there is a lack of bedding, a hollow is formed in the cubicle in which the cows do not like to lie [2]. Due to the softness of the straw, it is not possible to ensure sufficient thickness up to 200 mm without regular addition to the bedding, which was also confirmed in our study.

In practical breeding conditions the cubicles are not emptied of straw bedding, but fresh bedding is added to replace the soiled bedding. Ideally, the cubicle is re-bedded twice a day, always after the manure is cleared out from the end of the bed and from the corridors. On some farms the re-bedding procedure is performed only once a day with higher amounts of fresh bedding, but the corridors themselves are cleared out twice a day. In this case, excess bedding is cleared out of the cubicle into the corridors after re-bedding, and manure is produced with a high proportion of straw at the first clearing out. During the second clearing out, the straw is no longer taken out of the cubicles, and the manure is thin and almost free of straw [21].

Our results show that, in the case of classical straw bedding, its height in the bedded cubicles tends to fall below the floor barrier level, and the breeders have to resolve this by clearing out and adding new straw. Especially in straw-bedded cubicles, at the back (level with the cow's rear) the thickness of the bedding layer is often reduced, and there is an accumulation of dung and urine and an

abundance of bacteria, which in favorable conditions rapidly multiplies as demonstrated by increased numbers of TVC, CB, FCB, and FS in the control cubicles of our study. Conversely, we found that the RMS-improved composition maintained a stable level as well as reduced the numbers of TVC, CB, FS, and FCB after the first and second months of its use.

Bedding can be analyzed for a number of different bacteria, but not all bacteria will have an effect on udder health. A wide range of microorganisms can invade and infect the udder; however, staphylococci, coliforms, and streptococci are the most important in causing mastitis [22]. Therefore, these bacteria are the ones on which this study was focused. Coagulase-negative staphylococci (CNS), *S. aureus*, and *Str. sanquinis* were pathologically important in intramammary infection with positive CMT scores (score 1–3) in all monitored groups of cows. CNS was detected in almost 60% of all infected samples in the monitored groups. Although, some pathogens from the CNS group such as *S. chromogenes, S. warneri,* or *S. epidermidis* are historically considered to be of limited importance and are, therefore, often described as minor or environmental pathogens. In recent years, CNS have become increasingly important in udder infections probably because the prevalence of major pathogens has decreased [23].

Mastitis caused by CNS usually displays relatively mild clinical signs, and these bacteria can therefore affect milk quality for a long period before being noticed. In contrast, *Streptococcus uberis* is a widely distributed environmental pathogen causing more severe signs. The environmental pathogens are more difficult to eradicate due to their ubiquitous presence, and they remain a major challenge to the dairy industry [10]. They can be controlled by reducing exposure from the bedding environment with optimal composition [24].

The combination of RMS with limestone and straw, as well as their technological treatment, is a suitable alternative solution for reducing the infectious pressure from the environment. Improved bedding composition had a significant effect on the reduction of fecal contamination and on the number of mastitis events. As can be seen from Tables 1 and 2, the group of cows housed in cubicles after three months of alternative bedding use had a reduced occurrence of subclinical mastitis, which was reflected in a reduced number of infected quarters.

## 5. Conclusions

In our study, under farm conditions, we investigated the effect of an alternative composition of bedding used in a dairy farm on indicator microorganisms influencing the level of hygiene, in comparison to conventional straw bedding. Samples of bedding taken from monitored subsections and selected cubicles showed reduced TVC and CB counts in one-day-old alternative composition bedding as well as the first two months after it was laid. In addition to TVC and CB, decreased numbers of FCB and FS were recorded in one-day-old alternative bedding as well as in the first, second, and third months after use. In addition to reducing the number of microorganisms, the improved bedding had the stabilizing effect of keeping the bedding thickness up to the floor barrier (200 mm), which also had a beneficial effect on reducing the level of fecal contamination of the bedding.

By using the alternative composition bedding for a period of three months, the effect of reduced infection pressure from the environment was demonstrated, which resulted in an increased number of healthy quarters with negative CMT scores and a reduced incidence of subclinical mastitis in dairy cows.

However, RMS production and its addition together with other components (water, straw, and limestone) to bedding is not a standard procedure for dairy farmers in Slovakia. The use of RMS is relatively new, and the different compositions, materials, and production methods still have to be compared in combination with the presence of indicator microorganisms and the health status of cows.

**Author Contributions:** Conceptualization: F.Z. and S.O. Design of methodology: F.Z., N.S., and J.V. Samples and data analysis: G.G. and J.V. Resources: F.Z. and G.G. Writing and editing: F.Z. and S.O. All authors have read and agreed to the published version of the manuscript.

**Funding:** This research was funded by Slovak grants APVV no. SK-PL-18-0088, KEGA no. 006UVLF-4-2020, and VEGA no. 1-0529-19: The effect of environmental agents of mastitis in dairy cows and ewes on the production and degree of oxidative stress.

**Acknowledgments:** We would like to thank the owners of the Agricultural Cooperative Nová Ľubovňa (Slovakia) who allowed us to carry out the practical part of this study and collect samples at their premises.

**Conflicts of Interest:** The authors declare no conflicts of interest.

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
