# Peer review of "Effects of Using an Alternative Bedding Composition on the Levels of Indicator Microorganisms and Mammary Health in Dairy Farm Conditions"

_agriculture, doi:10.3390/agriculture10060245_

Round 1

Reviewer 1 Report

The topic treated is relevant and the study can potentially interest readers of Agriculture. However, the data collected are very limited and the study provides only little novel information. The introduction focused more on freestall dimensions/design rather than on microbiological issues, which represent the core of the paper. The methods are not adequately described as I struggled to understand clearly the figures in the results section. Statistical methods are not completely adequate as some correction for repeated measures should be used and reported. I would have appreciated the inclusion of another treatment with RMS only as bedding (rather than only one "control" with straw). This would have allowed a more comprehensive comparison with the bedding mixture proposed. Besides that, I think the main limitation of the study is the lack of any measure about the animals' status (i.e. cleanliness score) and mastitis risk. Reporting SCC of cows in the different groups would have increased the value of the study quite a lot. Authors mentioned that cows in the treatment groups remained cleaner, which indeed could have been a very interesting conclusion but is not supported by the results. Finally, I suggest rethinking the title of the paper. I would consider describing the bedding mixture proposed as "alternative" rather than "improved".

Author Response

Dear Reviewer

first of all, we thank you for your valuable and factual comments and advices to our manuscript. We have tried to improve the manuscript according to your suggestions.

Reviewer: The topic treated is relevant and the study can potentially interest readers of Agriculture. However, the data collected are very limited and the study provides only little novel information.

Comments: In our opinion, the study is innovative because it describes a new technological process for the production of improved bedding for dairy cows as well as the final stage of treatment. The available studies have reported the use of RMS only with the addition of limestone, straw, or sand but bedding was not further technologically treated in cubicles.

The study was extended by the obtained results of the occurrence of bacterial pathogens causing mastitis in dairy cows. At the same time as the bedding samples were taken, milk samples from housed cows in selected subsections with improved bedding were also taken, but due to lack of time, we did not manage to evaluate and include them in the study.

Reviewer: The introduction focused more on freestall dimensions/design rather than on microbiological issues, which represent the core of the paper.

Comments: The introduction has been changed and shortened with a focus on influencing the hygienic microbiological quality of bedding and its relationship to mastitis.

Reviewer: The methods are not adequately described as I struggled to understand clearly the figures in the results section.

Comments: The methods were correct and clearly described. The selection of individual sections and housed cows on bedding with improved composition was described in more detail. Cow milking and laboratory diagnosis of bacterial pathogens have also been described.

Reviewer: Statistical methods are not completely adequate as some correction for repeated measures should be used and reported.

Comments: The statistical methods were defined and described in more detail.

Reviewer: I would have appreciated the inclusion of another treatment with RMS only as bedding (rather than only one "control" with straw). This would have allowed a more comprehensive comparison with the bedding mixture proposed.

Comments: In each subsection were selected 9 cubicles (three cubicles in one row) and 30 cows. From each selected cubicle were taken 4 samples of bedding. A total of 180 bedding samples and 600 quarter milk samples were taken from all 5 subgroups. In our opinion, this is a sufficient number of samples to evaluate the hygienic quality of bedding and its effect on the occurrence of mastitis in the monitored farm.

Reviewer: Besides that, I think the main limitation of the study is the lack of any measure about the animals' status (i.e. cleanliness score) and mastitis risk. Reporting SCC of cows in the different groups would have increased the value of the study quite a lot. The authors mentioned that cows in the treatment groups remained cleaner, which indeed could have been a very interesting conclusion but is not supported by the results.

Comments: As mentioned in the introduction to our comments, the study was extended to the diagnosis of bacterial pathogens and the occurrence of mastitis in cows.

Reviewer: Finally, I suggest rethinking the title of the paper. I would consider describing the bedding mixture proposed as "alternative" rather than "improved".

Comments: We changed the word in the title of the paper from "improved" to "alternative".

Reviewer 2 Report

General comment:

The submitted paper for evaluation refers to effect of the improved bedding composition on the basis of recycled manure solids (RMS), used in dairy farms, on its technological property and mainly on the presence of selected bacteria. Since using of RMS bedding may pose a high health risk for an intrinsic dairy cow and thereafter also for consumers of the cow milk, I consider the subject of the study as interesting. The submitted study provides the advanced knowledge in the scientific field. However, the current status of the manuscript shows some lacks and flaws.            

Broad comment:

As the main contribution to current knowledge I consider the work findings regarding the prevalence of specific pathogenic bacteria in respective bedding materials used in the course of the assessed period of 3 months after their initial loading into cubicles. It is necessary, in my opinion, mainly to revise deeply Materials and methods part and to correct the use of particular technical (professional) English terms within the entire manuscript.    

Specific comments:

Abstract

Line (L) 19 – please, complete the statement concerning hygiene levels; hygiene levels of what…?

L. 26 - please, determine accurately the assessed period in the study.

L. 27 to 31 - please, rewrite the sentences in relation to usual scientific stylistics; mainly in “the second sentence”; now it is less understandable to readers.

L. 30 – delete the shorts FC and FS for the terms; they don’t appear again in abstract.

L. 35 – change to “bedded cubicles”       

Introduction

There is no hypothesis of your work mentioned here; please, put in the scientific hypothesis at the end of the chapter (somewhere before the stated aim of the study).

Fig. 1 – please revise and complete the diagram stated; missing arrow for bed management, storage and handling of what…?, particle size of what…?, nutrient composition of what …?

Fig. 2 – I consider it as redundant.      

L. 41 – change to …”a system of housing which”…

L. 68 – revise the sentence in relation to the EN grammar and stylistics.

Material and methods:

The experimental arrangement raises some doubts or lacks. This chapter must be deeply revised and partially reworked. Use correct technical (professional) terms here, mainly for housing technology assessed, proceeding of RMS and performing the improved bedding composition.

Fig. 4 – double marked… which is the figure 3 and figure 4 …? Moreover, you refer to the Fig. 4  regarding a two-row section of the stable in your assessment, whereas in displayed housing module designed by Samer et al. (2009) is in fact illustrated a three-row section (pen) for dairy cows….? How many rows in the section were in the assessed stable? Revise it properly in the M α M!

There is missing the experimental design now (!) … It must be correctly stated somewhere in this part!; mainly as for the assessed types of bedding, timing of bedding sampling for microbiol. examination etc.     

Tab. 1 -  I consider it as redundant.      

L. 88 – change to … “in the farm”…

L. 90 – change to… “feeding table”..

L. 91 – please, provide more detailed correct information of the section subdivision

L. 91 – provide the correct number of rows in sections assessed

L. 96 – change to “Bedded Cubicles”

L. 103 – please, use a better term for “transverse withers barrier”

L. 119-117 – change to … “feeding car”…

L. 131 – change to “Sampling of Bedded Cubicles”

L. 132-133 – rewrite the sentence in relation to better stylistics

L. 133 Determine precisely the sampling of bedding in the course of your assessment here.

L. 137 – change to …“manure is cleaned out”…; revise it in the whole manuscript

L.141 to 143 – the sentence doesn’t make sense now; please correct it

L. 156 – explain the short “CPM”

Results

Results section provide adequate results presentation of the conducted present study.

L. 170 – correct to – …“to the conventional straw bedding”…

Fig. 10 – correct marking to “C–fresh straw” for particular column  

Discussion:

Discussion as a chapter is sufficiently extensive.

L. 229-230 – please, particularize the stated ratio of 3:1; (RMS to ground limestone…?)

Conclusion

The substantial findings are stated here, that rising from the obtained results of the assessment. 

L. 286 – please, correct to …”a cleaner lower part of body and”…

L. 288 – please, state here the specific country (Slovak Republic …?)

Author Response

Dear Reviewer

first of all, we thank you for your valuable and factual comments and advices to our manuscript. We have tried to improve the manuscript according to your suggestions.

General comment:

According to recommendations one of the reviewers, the study was extended by the obtained results of the occurrence of bacterial pathogens causing mastitis in dairy cows. At the same time as the bedding samples were taken, milk samples from housed cows in selected subsections with improved bedding were also taken, but due to lack of time, we did not manage to evaluate and include them in the study.

Specific comments and corrections:

Abstract

Line (L) 19 – please, complete the statement concerning hygiene levels; hygiene levels of what…?

Comments: It has been added to the text "influencing hygiene level in cubicle floors and occurrence of mastitis in dairy cows".

26 - please, determine accurately the assessed period in the study.

Comments: The assessed period in the study was clearly described according to the time intervals of bedding treatment into individual subsections for three months.

27 to 31 - please, rewrite the sentences in relation to usual scientific stylistics; mainly in “the second sentence”; now it is less understandable to readers.

Comments: The scientific intent of the study as well as the results were stylistically reworked in the abstract.

30 – delete the shorts FC and FS for the terms; they don’t appear again in abstract.

Comments: The shorts FC and FS have been removed from the abstract.

35 – change to “bedded cubicles” 

 Comments:  Context changed to “bedded cubicles”.

Introduction

There is no hypothesis of your work mentioned here; please, put in the scientific hypothesis at the end of the chapter (somewhere before the stated aim of the study).

Comments:  According to the recommendations, a hypothesis was added to the text.

Fig. 1 – please revise and complete the diagram stated; missing arrow for bed management, storage and handling of what…?, particle size of what…?, nutrient composition of what …?

Comments:  The diagram was revised and completed.

Fig. 2 – I consider it as redundant.    

Comments: We delete Fig. 2.

41 – change to …”a system of housing which”…

Comments: Context changed to …”a system of housing which”…

68 – revise the sentence in relation to the EN grammar and stylistics.

Comments: The context changed to …”and increases the risk of mammary gland inflammation”…

Material and methods:

The experimental arrangement raises some doubts or lacks. This chapter must be deeply revised and partially reworked. Use correct technical (professional) terms here, mainly for housing technology assessed, proceeding of RMS and performing the improved bedding composition.

Comments: The methods were correct and clearly described. The selection of individual sections and  housed cows on bedding with improved composition was described in more detail. Cow milking and laboratory diagnosis of bacterial pathogens have also been described.

Fig. 4 – double marked… which is the figure 3 and figure 4 …? Moreover, you refer to the Fig. 4  regarding a two-row section of the stable in your assessment, whereas in displayed housing module designed by Samer et al. (2009) is in fact illustrated a three-row section (pen) for dairy cows….? How many rows in the section were in the assessed stable? Revise it properly in the M α M!

Comments: Some figures have been deleted. Figures are corrected and renumbered. The picture according to Samer et al. (2009) was clearly described in the methods of the study. Each section is divided into three rows with 42 cubicles.

There is missing the experimental design now (!) … It must be correctly stated somewhere in this part!; mainly as for the assessed types of bedding, the timing of bedding sampling for microbiol. examination etc.    

 Comments: In methods is added description of experimental design and sampling of improved bedding and milk from housed cows.

Tab. 1 -  I consider it as redundant.   

 Comments: We delete table 2.

88 – change to … “in the farm”…

90 – change to… “feeding table”..

 Comments: The L. 88 and 90 were changed according to your recommendations.

91 – please, provide more detailed correct information of the section subdivision

91 – provide the correct number of rows in sections assessed

Comments: After corrections, the sections and subsections are clearly described in the text.

96 – change to “Bedded Cubicles”

Comments: Context changed to “Bedded Cubicles”.

103 – please, use a better term for “transverse withers barrier”

Comments: Context changed to “withers barrier”

119-117 – change to … “feeding car”…

Comments: Context changed to … “feeding car”…

131 – change to “Sampling of Bedded Cubicles”

Comments: Context changed to “Sampling of Bedded Cubicles”

132-133 – rewrite the sentence in relation to better stylistics

Comments: The stylistics was changed in the text.

133 Determine precisely the sampling of bedding in the course of your assessment here.

Comments: Bedding samples for microbiological examination were clearly described in the text.

137 – change to …“manure is cleaned out”…; revise it in the whole manuscript

Comments: Context changed to  …“manure is cleaned out”…

L.141 to 143 – the sentence doesn’t make sense now; please correct it

Comments: Context has been correct for better understanding.

156 – explain the short “CPM”

Comments: This part of context was changed a clearly described.

Results

Results section provide adequate results presentation of the conducted present study.

170 – correct to – …“to the conventional straw bedding”…

Comments: Context changed to …“twith conventional straw bedding”…

Fig. 10 – correct marking to “C–fresh straw” for particular column  

Comments: The fig 10 was changed according to the recommendation of one of the reviewers.

Discussion:

Discussion as a chapter is sufficiently extensive.

229-230 – please, particularize the stated ratio of 3:1; (RMS to ground limestone…?)

Comments: The ratio of RMS and ground limestone is supplemented in the text

Conclusion

The substantial findings are stated here, that rising from the obtained results of the assessment. 

286 – please, correct to …”a cleaner lower part of body and”…

Comments: Context changed to …”a cleaner lower part of body and”…

288 – please, state here the specific country (Slovak Republic …?)

Comments: The country was added in the text.

Reviewer 3 Report

This paper provided detailed technical detail about the dairy cattle in-house management for bedding arrangement options from the perspective of improving hygiene and animal health.

  1. Line 202, the note of Figure indicated that there is a treatment for "C-Fresh Straw" but such option was not shown on the Figure.
  2. Lines 95, 114, 122, 130,Spelling errors need to be fixed: "Foto" should be "Photo"
  3. Please explain the sampling design, sample size and timing.Would the outside environment affect the results of the experiment?
  4. The paper used a current farm to experiment on the different bedding option. However, it is not clear how representative such setting is?

Author Response

Dear Reviewer

first of all, we thank you for your valuable and factual comments and advices to our manuscript. We have tried to improve the manuscript according to your suggestions.

General comment:

According to recommendations one of the reviewers, the study was extended by the obtained results of the occurrence of bacterial pathogens causing mastitis in dairy cows. At the same time as the bedding samples were taken, milk samples from housed cows in selected subsections with improved bedding were also taken, but due to lack of time, we did not manage to evaluate and include them in the study.

Specific comments:

Line 202, the note of Figure indicated that there is a treatment for "C-Fresh Straw" but such option was not shown on the Figure.

Comments: The fig 10 was changed according to the recommendation of one of the reviewers.

Lines 95, 114, 122, 130, Spelling errors need to be fixed: "Foto" should be "Photo"

Comments: Context changed to  "Photo".

Please explain the sampling design, sample size and timing. Would the outside environment affect the results of the experiment?

Comments: The methods were correct and clearly described. The selection of individual sections and housed cows on bedding with improved composition was described in more detail. Cow milking and laboratory diagnosis of bacterial pathogens have also been described.

The paper used a current farm to experiment on the different bedding option. However, it is not clear how representative such setting is?

Comments: The study was performed in order to increase the hygienic standard in dairy farming. The results are applicable to all dairy farms using bedding from RMS.

Round 2

Reviewer 1 Report

Dear Authors,

thanks for having expanded the study with udder health data. I think it is now much more interesting. That said, I still struggle to understand the methods used. In particular, I do not understand when the bedding samples and udder/milk observations have been collected for the different groups. Were observations collected simultaneously for all groups? In this case, if I understood correctly, the sections "IB after 1M", "IB after 2M" and "IB after 3M" were bedded 1, 2 and 3 months before the actual observation period, respectively. Is that correct? In any case, I strongly recommend improving the description of the treatments (Section 2.3). In the responses, you stated that "A total of 180 bedding samples and 600 quarter milk samples were taken from all 5 subgroups". This is very relevant but it is not reported in the manuscript. So please, expand the description of the sampling procedures for both bedding and udder/milk including the time of collection and the number of samples (sections 2.4 and 2.5).

All figures need labels for both x and y axes.

In Table 1 and 2 and in Figure 10, significant differences are marked with "*". That's unclear as the reader cannot appreciate which groups are significantly different. Can you use superscript letters as in all other Figures?

In the conclusions (line 435) you stated that cows can be kept cleaner by using this bedding mixture. However, there are no results to support this statement as cow cleanliness was not assessed. Please remove this statement from the conclusions.

I'm glad to see the bedding mixture studied is now described as "alternative" in the title. However, the adjective "improved" is still used extensively in the text. Consider using the same adjective throughout the manuscript.

Finally, I strongly recommend the editing of the English language before resubmission (ideally by professional English editing service).

Author Response

Dear Reviewer

thanks for reviewing the article again. We are glad that all our corrections from the first round of article review have been accepted.

Specific comments and corrections:

Reviewer: That said, I still struggle to understand the methods used. In particular, I do not understand when the bedding samples and udder/milk observations have been collected for the different groups. Were observations collected simultaneously for all groups? In this case, if I understood correctly, the sections "IB after 1M", "IB after 2M" and "IB after 3M" were bedded 1, 2 and 3 months before the actual observation period, respectively. Is that correct? In any case, I strongly recommend improving the description of the treatments (Section 2.3).

Comments: In the methods, section 2.3 is renumbered to section 2.5 and is clearly described experimental groups selection as well as observation period. You understood correctly that all selected groups were monitored simultaneously. The sections "IB after 1M", "IB after 2M" and "IB after 3M" were bedded 1-3 months before the actual observation period, respectively.

Reviewer: In the responses, you stated that "A total of 180 bedding samples and 600 quarter milk samples were taken from all 5 subgroups". This is very relevant but it is not reported in the manuscript. So please, expand the description of the sampling procedures for both bedding and udder/milk including the time of collection and the number of samples (sections 2.4 and 2.5).

Comments: In the methods, sections 2.4 and 2.5 are renumbered to sections 2.6 and 2.7, and are clearly described. The bedding and milk samples were taken simultaneously for all groups. As mentioned earlier, sections with alternative bedding were bedded 1-3 months before the actual observation period.

Reviewer: All figures need labels for both x and y axes.

Comments: Labels for both x and y axes have been added to all figures.

Reviewer: In Tables 1 and 2 and in Figure 10, significant differences are marked with "*". That's unclear as the reader cannot appreciate which groups are significantly different. Can you use superscript letters as in all other Figures?

Comments: We changed significantly differences marked with "*" to superscripts letters in tables 1 and 2 as well as and fig. 10.

Reviewer: In the conclusions (line 435) you stated that cows can be kept cleaner by using this bedding mixture. However, there are no results to support this statement as cow cleanliness was not assessed. Please remove this statement from the conclusions.

Comments: We removed the part about the purity of cows and changed the text in conclusion.

Reviewer: I'm glad to see the bedding mixture studied is now described as "alternative" in the title. However, the adjective "improved" is still used extensively in the text. Consider using the same adjective throughout the manuscript.

Comments: We tried to change the term "improved" to an "alternative" in the text.

Reviewer: Finally, I strongly recommend the editing of the English language before resubmission (ideally by professional English editing service).

Comments: After the second round of revision we will send an article for professional English editing to Agriculture service.

Reviewer 2 Report

General comments

The authors performed substantial revision of their work in revised version that responds to my foregoing comments.

Specific comments:

Please, harmonize the marking of revised figures and their references through the entire textual part of manuscript.

Line 202 – 203 - please, revise …”of white blood cells or leukocytes”…???

Line 220 – change to …“zeolite should ensure hygienic”…      

Author Response

Dear Reviewer

thanks for reviewing the article again. We are glad that all our corrections from the first round of article review have been accepted. After the second round of revision, we will send an article for professional English editing to Agriculture service.

Specific comments:

Reviewer: Please, harmonize the marking of revised figures and their references through the entire textual part of manuscript.

Comments: We controlled markings of revised figures again. We renumbered the sections in methods. In table 1 and 2 were changed significantly differences marked with "*" to superscripts letters.

Reviewer: Line 202 – 203 - please, revise …”of white blood cells or leukocytes”…???

Comments: Text was revise …”of white blood cells and this causes an increase in the somatic cell count (SCC).

Reviewer: Line 220 – change to …“zeolite should ensure hygienic”…      

Comments: Text was changed according to your recommendations.